# RADAR-IoT: An Open-Source, Interoperable, and Extensible IoT Gateway Framework for Health Research

**DOI:** 10.3390/s24144614

**Published:** 2024-07-16

**Authors:** Yatharth Ranjan, Jiangeng Chang, Heet Sankesara, Pauline Conde, Zulqarnain Rashid, Richard J. B. Dobson, Amos Folarin

**Affiliations:** 1Biostatistics and Health Informatics, Institute of Psychiatry, Psychology & Neuroscience, King’s College London, London SE5 8AF, UK; yatharth.ranjan@kcl.ac.uk (Y.R.); heet.sankesara@kcl.ac.uk (H.S.); pauline.conde@kcl.ac.uk (P.C.); zulqarnain.rashid@kcl.ac.uk (Z.R.); richard.j.dobson@kcl.ac.uk (R.J.B.D.); 2Saw Swee Hock School of Public Health, National University of Singapore, Singapore 119077, Singapore; e0950658@u.nus.edu

**Keywords:** IoT, gateway, health, research, RADAR-base, mHealth, remote monitoring, extensible

## Abstract

IoT sensors offer a wide range of sensing capabilities, many of which have potential health applications. Existing solutions for IoT in healthcare have notable limitations, such as closed-source, limited I/O protocols, limited cloud platform support, and missing specific functionality for health use cases. Developing an open-source internet of things (IoT) gateway solution that addresses these limitations and provides reliability, broad applicability, and utility is highly desirable. Combining a wide range of sensor data streams from IoT devices with ambulatory mHealth data would open up the potential to provide a detailed 360-degree view of the relationship between patient physiology, behavior, and environment. We have developed RADAR-IoT as an open-source IoT gateway framework, to harness this potential. It aims to connect multiple IoT devices at the edge, perform limited on-device data processing and analysis, and integrate with cloud-based mobile health platforms, such as RADAR-base, enabling real-time data processing. We also present a proof-of-concept data collection from this framework, using prototype hardware in two locations. The RADAR-IoT framework, combined with the RADAR-base mHealth platform, provides a comprehensive view of a user’s health and environment by integrating static IoT sensors and wearable devices. Despite its current limitations, it offers a promising open-source solution for health research, with potential applications in managing infection control, monitoring chronic pulmonary disorders, and assisting patients with impaired motor control or cognitive ability.

## 1. Introduction

In the rapidly evolving world of digital health, integrating mobile health (mHealth) and internet of things (IoT) platforms is revolutionizing how we collect, analyze, and utilize health data. This paper delves into the opportunities and challenges this technological convergence presents and proposes our innovative solution to addressing the identified limitations. Given these challenges, our work was motivated by the need for an open-source, mHealth-focused IoT platform that seamlessly integrates and processes data from diverse sources. Our goal is to bridge the gap between the burgeoning IoT market and specific health research needs, thereby unlocking the IoT’s full potential for advancing health research.

The landscape is witnessing a significant expansion in modern sensing capabilities and network-connected devices. This growth presents an opportunity to collect large-scale health data continuously and remotely, facilitating timely analysis for identifying and monitoring individuals’ health status [1]. Numerous established and emerging mobile health (mHealth) and internet of things (IoT) platforms exist within the market. These platforms have gained increasing importance, due to technological advancements and the widespread integration of wearables into daily life. As a result, unprecedented possibilities (e.g., home-based and ambulatory monitoring and just-in-time interventions) have emerged in research and clinical applications [2].

The internet of things is a network of physical objects, “things”, comprised of sensors, software, and other technologies, to interact data with other devices and systems over the internet. The internet of things represents a broad category of devices and sensors with the common attribute of network connectivity; they form a superset that includes wearable devices and smartphones [3]. An IoT gateway device acts as a hub. It interconnects all IoT devices at a location (called “at-edge”, i.e., near the source of the data) to the cloud-based platforms and IoT devices in other locations [4]. Its functionality may also include rudimentary processing of data and security- and privacy-oriented access. This scenario has become increasingly realistic and exciting with recent developments, such as neural processing units (NPUs), 64-bit CPUs at-edge, and advanced artificial intelligence.

The main contribution of this paper is implementing a flexible IoT gateway framework, RADAR-IoT, directed towards improving health research. By incorporating IoT capabilities into the open-source RADAR-base mHealth platform, we aim to overcome the limitations of wearable device integration and to unlock a broader spectrum of health data collection possibilities, thereby enhancing the depth and breadth of insights derived from the collected data. This paper presents a comparative analysis of IoT frameworks and the motivations behind the proposed solution. This proposed IoT framework focuses on extensibility, modularity, interoperability, performance, and utility in health research, among other vital features.

The paper is divided into six sections, as follows: Section 2 discusses the background of the IoT in the health domain, the RADAR-base platform and its applications, related work, and the design requirements for the IoT gateway. Section 3 describes the architecture of our proposed IoT gateway framework, as informed by the requirements in Section 2; this section also provides details about the salient features of the framework, including performance, security, and privacy. Section 4 demonstrates the proof of concept (PoC) for RADAR-IoT, to evaluate the framework in real-world settings. Section 5 discusses the contribution of our proposed solution to the IoT in health research and its potential applications. Section 6 provides the conclusion of the overall paper, including limitations and future work.

### Motivation

The internet of things (IoT) sector of the healthcare market is projected to reach USD 534.3 billion by 2025, expanding at a compound annual growth rate (CAGR) of 19.9% [5], as shown in Figure 1. In 2018, the market was valued at USD 147.1 billion. This significant growth can be attributed to the proliferation of wearables and IoT-enabled devices in daily life, such as smart virtual assistants and smart lighting, as well as the emergence of cloud computing, which enables the management and deployment of millions of connected IoT devices. However, the uptake of these technologies in the medical/health research domain is more attenuated, with more than 50% of doctors expressing concerns over IoT devices’ reliability [6]. These concerns arise because the IoT data are often unprocessed, non-standardized, and lack actionable or valuable intelligence. Integrating data from multiple data sources with an mHealth platform capable of analyzing real-time data at both the network edge and in the cloud has the potential to generate actionable information. The system can achieve a detailed 360-degree view of the patient’s physiology, behavior, and environment by leveraging this confluence of data and edge-processing capability, combined with appropriate validation.

While a wide range of IoT platforms is available in the market, most are commercial or closed-source, and even open-sourced platforms are typically consumer-oriented, such as those designed for controlling smart home devices. However, there is a need for open-source mHealth platforms that allow for interconnected IoT devices and offer a standard interface for collecting, aggregating, and analyzing real-time data from multiple sources. These sources include wearables, IoT sensors, mobile devices, and medical devices, which are essential for health research and more challenging to translate to clinical domains.

## 2. Background

Addressing challenges in health research and the limitations of existing IoT frameworks is necessary. Specific requirements, such as extensibility, modularity, reliability, security, and efficiency, should guide the design and development of future solutions, along with the support of health platforms, such as RADAR-base. These solutions underscore the IoT’s potential in health research and present a viable solution for researchers in the field.

### 2.1. Internet of Things

Surveys on the IoT in healthcare [7,8] reveal that most IoT platforms suffer from limitations, such as limited sensor support, closed-source systems, limited I/O protocols, no proper cloud platform support, no demonstrable proof of concept, and limited extensibility. The limitations identified in these surveys have significant practical implications for health research. Open-source software promotes transparency, community collaboration, and potential cost-effectiveness and could bolster applications in health research. The limited sensor support and closed-source systems can hinder the integration of diverse medical devices and data sources, potentially limiting the scope and accuracy of health research. Additionally, the absence of proper cloud platform support and limited I/O protocols may impede the seamless collection, storage, and analysis of health data, which is crucial for conducting comprehensive research.

Moreover, the need for verifiable proof of concept and limited extensibility could restrict the scalability and adaptability of IoT platforms in health research, potentially constraining innovation and advancements in the field. Addressing these limitations is essential to ensure the robustness and effectiveness of IoT applications in healthcare research. To address these issues, we have designed an open-source internet of things (IoT) gateway framework that can interconnect multiple IoT devices on the edge, perform rudimentary analysis, and integrate well with established mHealth platforms in the cloud. Finally, we evaluated these capabilities in a proof-of-concept deployment.

### 2.2. RADAR-Base Platform

RADAR-base [9] is an open-source platform that integrates data streams from various wearables and mobile technologies, to collect sensor data in real time and to store, manage, and share the collected data, knowledge, and insights with researchers for analysis and actionable intelligence. As shown in Figure 2, it supports both passive and active data collection through two applications, pRMT and aRMT, which monitor movement, location, audio, calls, texts, and app usage, and it includes questionnaires, to gather patient information. RADAR-base has been and is currently being used in various research studies. Refs. [10,11,12,13,14] focusing on personal sensing. However, the vendor availability of SDKs and REST APIs limits the platform’s integration with wearable devices. The proposed framework in this paper extends the platform to include IoT sensing capability, expanding the range of use cases.

### 2.3. Related Work

We compare various IoT gateway frameworks in Table 1. RADAR-IoT [15] takes inspiration from such frameworks for its design and does not mean to replace them. Instead, it has the potential to connect to them, to collect or pass data to them, hence making use of the existing functionalities of these platforms. While generalized platforms like these are good for broader use cases, they do not shine for specific use cases, such as health research and remote monitoring of patients. Most of the platforms typically use a platform-specific message bus, like D-Bus. Still, the benefits of having an external industry-standard message broker are manifold, such as (i) event-driven programming and analysis, (ii) language and platform-agnostic publishers and subscribers, (iii) performance, (iv) extensibility, and (v) interoperability. Some platforms are primarily cloud-based (these can only function with a connection to their respective cloud servers); this has clear limitations in privacy-constrained environments, such as hospitals, where a local network is preferred. The existing IoT gateways shown in Table 1 are primarily intended for commercial or industrial use. Some health and research organizations provide gateways as physical devices [16], limiting their adoption and control. Kesavan et al. [17] proposed a gateway framework for smart health; while well-designed, it is limited to a few body-attached sensors, supports only Bluetooth and Wi-Fi, and provides limited information on cloud platform integration.

The IoT ecosystem is very diverse, with a wide range of hardware manufacturers, and application areas and standards have thus far been challenging to realize. The large-scale network, IoT sensors heterogeneity, and the volume of IoT data pose challenges for application development [29]. Middleware can offer standard services for applications and support interoperability within applications and services running on these hub devices. RADAR-IoT aims to be a middleware to overcome such challenges.

The proposed IoT framework’s (RADAR-IoT) primary objective is to augment the existing RADAR-base mHealth platform [9] by incorporating IoT sensing capabilities. RADAR-base presently focuses on ambulatory mHealth data from phones and wearables; RADAR-IoT will provide a broader range of sensory insights into the complex interplay between patients’ physiology, behavior, and the surrounding environment. The addition of RADAR-IoT will lead to more informed healthcare decisions and personalized interventions, e.g., exposure to poor air quality in pulmonary disorders, such as COPD.

### 2.4. Design Requirements

To meet the requirements discussed previously, we propose an IoT edge framework. The design ideas for the proposed IoT edge framework are depicted in Figure 3. Essentially, the design was informed based on the following requirements:

Extensibility and flexibility: allow integration of any sensor, source, and destination.Modularity: allow for services/components to communicate through a standard API, allowing them to be augmented or replaced by custom implementations, facilitating re-usability.Support a publish–subscribe design pattern [30]: allow local-level interoperability, reliability, and scalability at the edge.Allow data to be consumed, processed, and uploaded flexibly.Operate on low-power and resource-edge devices (like Raspberry Pi zero [31]).Observability: the system should allow continuous monitoring from various sensors with different hardware interfaces with data standardization.Reliability: support high-fidelity data collection at scale and throughput.Data analysis: including artificial intelligence (AI) applications and machine learning (ML) at the edge.Enable remote deployment and orchestration of services at scale.Provide for interoperability with other IoT devices complying with a subset of the OGC SensorThings API standard [32].Support service and machine discovery at the edge.Designed using modern technologies, to obtain best-in-class features and performance.

## 3. Proposed Solution

We designed RADAR-IoT (RADAR-base—internet of things) as an IoT gateway [4] framework for connecting a wide variety of sensors and flexible IoT sensor data processing and uploading. RADAR-IoT also supports uploading to the RADAR-base platform and InfluxDb [33] (local or cloud). It also provides real-time visualization on Grafana [34] dashboards and can be extended to include more cloud destinations. All the entities are based on abstract base components and provide extensions for adding more data-processing variants and sensors. The use of an external industry standard publish–subscribe [30] platform like Redis Pub/Sub [35] or MQTT [36] makes the RADAR-IoT framework language and platform agnostic.

### 3.1. Architecture Overview

The architecture of the RADAR-IoT framework, as shown in Figure 4, illustrates its components and data flows from left to right, culminating in data transmission to external services. The framework is flexible and compatible with various IoT sensors (e.g., environment sensors, video, and audio) through standard input/output protocols (like I2C, GPIO, Serial, and more). Data are validated using a schema and converted into a format suitable for downstream components. These messages are then forwarded to or published on the publisher–subscriber platform, where multiple consumers can access them. The consumer reads these messages and uploads them to external services, such as the dashboard and the RADAR-base platform [9]. Once data are published, they can be used in diverse ways, including cloud transmission, on-device data analysis or processing, and Node-RED [37] flows (see below).

### 3.2. Publish–Subscribe Broker or Message Queue

RADAR-IoT uses a Publish–Subscribe (pub–sub) broker [30] as a message queue for publishing and consuming sensor data, providing several benefits. Firstly, an external and widely used pub–sub system (such as Redis [35], Kafka [38], or MQTT [36]) enhances reliability compared to a custom internal message bus by allowing direct data syncing to the cloud running the same pub–sub broker. Secondly, the framework enables bi-directional production and consumption of data between the data consumer and external applications or services, enabling multiple types of publishers, such as broadcasting data from one device to another anywhere on the same network. Thirdly, event-driven programming and analysis are supported. Lastly, publishers and subscribers are language- and platform-agnostic.

### 3.3. OGC Sensor Things API

The Open Geospatial Consortium OGC SensorThings API provides an open, geospatial-enabled, and unified way to interconnect internet of things (IoT) devices, data, and applications over the Web [32]. We used a subset of specifications from the OGC API specification to design the RADAR-IoT framework.

The channel or topic names in the publish–subscribe system provide a standard to specify the task, hence providing a standardized way to publish and consume data. This approach can expose canonically named endpoints to control and configure the sensors. For instance, for a sensor with <sensor-name>, we use the following format for various endpoints:/sensor/<sensor-name>/data-stream/sensor/<sensor-name>/error/sensor/<sensor-name>/control/sensor/<sensor-name>/config

We have exposed an endpoint that contains comprehensive metadata from the sensors in operation, making the system self-describing.

### 3.4. State Machine

A state machine, also known as a finite state machine, is a mathematical model that describes the discrete behavior of systems. It consists of a set of states, transitions between those states, and actions. In a state machine, the system’s elements are always in one of a finite number of states, and events or conditions trigger the transitions between those states. Incorporating a state machine [39] into the sensor abstractions, to capture and monitor sensor life cycle events, confers enhanced visibility and insight into the framework and sensors, enabling more resilient fault isolation, diagnosis, and efficient resolution. The resulting event logs of a particular state are also disseminated to the pub–sub system, thereby enabling downstream and upstream consumers to have full awareness of the system state and to leverage this information responsively for a wide range of applications.

### 3.5. Abstraction

The RADAR-IoT framework leverages interfaces and abstract classes. Adopting object-oriented programming principles [40] grants the system enhanced flexibility and extensibility. Among the specific advantages are the following:The majority of the components are abstracted and re-usable. The Sensor abstract class is the fundamental unit for collecting sensor data, with all sensor implementations required to extend this class or one of its subclasses.The framework facilitates the creation of new abstractions based on existing ones. For instance, by adding a group of sensors that interact coherently through a sensor HAT or interface (like GrovePi HAT [41]) we can extend the Sensor class to add HAT-specific implementations (like GrovePiSensor), allowing us to extend abstract functions in further sensor-specific subclasses (like GrovePiTemperatureSensor). This approach provides a modular and extensible design, where the core Sensor class handles standard functionality. At the same time, sensor-specific implementations are encapsulated in subclasses, promoting code re-use, maintainability, and flexibility in accommodating diverse sensor types within the system.This approach enables reliable incremental testing. Since the abstractions have already been tested, only the specific extensions must be unit-tested. Incremental testing enables targeted and efficient testing, as we only need to verify the new implementation details rather than re-testing the entire system.

### 3.6. Security and Privacy

Security and privacy are paramount concerns when collecting health and environmental data, particularly given IoT systems’ substantial number of endpoints. To address these issues, RADAR-IoT incorporates several measures:Secure Data Collection: A secure channel, accessible only through hardware interfaces, gathers sensor data. Packaging the hardware in a robust, tamper-proof case can further ensure security.Authentication and Authorization: After the data are gathered from the sensor, they are transmitted to the publish–subscribe broker. The broker runs in a trusted environment and is inaccessible through the network. The broker can be configured with solid security, ensuring publishers and subscribers authenticate before accessing or submitting data, and preventing unauthorized access. The security features will vary based on the publish–subscribe system used. Redis Pub/Sub [35] and MQTT [36] brokers support password authentication and access control lists (ACLs) authorization methods. MQTT additionally also supports encryption and auditing. The pub–sub channels can be secured using industry-standard encryption protocols, such as TLS/SSL, ensuring the data are protected from eavesdropping or tampering during transmission.Pseudonymization: All data are associated with pseudonymized identifiers, to safeguard privacy and prevent user or environment identification. This process ensures that the data cannot be directly linked to a specific user or environment. The user’s private identifiable information (PII) is kept separately in a secure environment with limited access and decoupled from the sensor data.On-Device Feature Extraction: When identifiable data, such as audio, are involved, edge devices perform feature extraction of non-reversible features, which can be shared instead of the original data. This feature extraction capability is integrated into the RADAR-IoT framework as a Node-RED module, ensuring that sensitive raw data never leave the device.Cloud Security: When uploading data to the cloud, RADAR-IoT conforms to the standard security mechanisms of the target systems. For instance, RADAR-IoT employs the OAuth 2.0 [42] industry standard for securing data at REST when uploading data to the RADAR-base platform, along with TLS/SSL encryption. These measures ensure that the data are protected both in transit and at rest and that only authorized applications can access the data.

By implementing these measures, RADAR-IoT provides a robust security and privacy framework, to protect sensitive health and environmental data from potential threats.

### 3.7. Performance

RADAR-IoT’s performance is optimized through several key strategies, including multi-process scheduling, efficient sensor data processing, queue use, robust exception handling, and fault isolation.

Multi-Process Scheduling: RADAR-IoT employs a multi-process scheduling approach to handling multiple tasks concurrently. This approach allows the system to process data from multiple sensors simultaneously, significantly improving the system’s overall throughput. Each process is ensured adequate CPU time by the scheduling algorithm, preventing any process from monopolizing the system resources.Processing of Sensor Data: The system efficiently processes large volumes of sensor data. It employs efficient data structures and algorithms, minimizing the computational overhead. The system also uses caching techniques to store frequently accessed data in memory, reducing the need for expensive disk I/O operations.Use of Queues: RADAR-IoT uses queues to manage the data flow after receiving the data from the sensors. Queues ensure that even if a component is temporarily unable to process incoming data (e.g., if the pub–sub system is busy or down), the data are not lost but are instead queued for later processing. This queue-based approach enables ordered data processing and helps to smooth out variations in data arrival rates, preventing system overloads.Robust Exception Handling: The system includes robust exception-handling mechanisms, to ensure that it can recover gracefully from errors. When an exception occurs, the system logs the error for later analysis then either attempts to correct the error or skips the problematic operation, depending on the nature of the error. This ensures that a single error does not cause the entire system to fail.Fault Isolation: We designed RADAR-IoT with fault isolation in mind. Each system component runs in its own process space, so if one component fails, it does not affect the others. Each component can be easily identified and diagnosed for faults with this design, as every part is testable and debuggable independently.

By implementing these strategies, RADAR-IoT ensures high performance and reliability, even when dealing with large volumes of sensor data and high system activity levels.

### 3.8. Data Typing and Schematization

Avro schemas [43] are a crucial tool for data typing, validation, and standardization. Before publishing sensor data to the pub–sub broker, the system validates and converts it, using Avro schemas. This process ensures consistent data representation and facilitates schema evolution and backward compatibility. The converter can read schemas from various sources, including local files, GitHub-hosted files, and the Confluent Schema Registry component [44]. The RADAR-base platform accepts Avro-formatted data, hence streamlining data upload.

### 3.9. Data Processing and Analysis

The RADAR-IoT framework is designed to handle real-time sensor data and provides an extensible module for data processing and analysis that can support various methods. Here is a breakdown of some of the currently supported methods:tinyML [45]: This is a paradigm that enables AI and machine learning (ML) in low-power and resource-constrained edge devices. The rationale behind using tinyML is to bring the power of ML to IoT devices, which are typically constrained by power and computational resources. This allows for real-time, on-device processing and analysis of sensor data, reducing the need for data transmission and preventing privacy leaks.Data aggregations and transformations: These are essential data processing methods that are used to prepare and clean the sensor data for further analysis. Aggregation methods might include calculating specific features over precise time intervals. Transformation methods might include normalization and scaling. These methods are essential for dealing with real-time sensor data, which can be noisy and come in large volumes.Node-RED [37]: Node-RED is a programming tool for wiring together hardware devices, APIs, and online services in new and exciting ways [46,47,48,49]. It provides a browser-based flow editor that makes it easy to wire together flows, using the wide range of nodes in the palette. Flows can then be deployed to the runtime in a single click. Using Node-RED allows for easy and flexible data analytics, using a no-code web interface, making it accessible to users without a coding background. Figure 5 shows an example data processing pipeline using Node-RED for audio feature extraction.RADAR-base cloud platform [9]: For more advanced analysis that requires high computational resources, the framework can delegate the analysis to cloud platforms. This allows for using more complex ML models and algorithms that may not be feasible to run on edge devices, due to resource constraints. Moreover, these cloud platforms can also integrate and analyze data from various sources, including wearables and mobile phones. This multi-source data integration can provide a more comprehensive view of the situation, thereby improving the quality of the analysis and the accuracy of the results.

In summary, the rationale behind the chosen methods and models is to provide a flexible, scalable, and accessible framework for real-time sensor data analysis, capable of running both on resource-constrained edge devices and on powerful cloud platforms. The choice of methods allows for a wide range of analysis tasks, from basic data cleaning and preparation to advanced ML models. The use of tools like Node-RED also makes the framework accessible to non-programmers. More advanced analysis methods are planned for future implementation, which will further enhance the capabilities of the RADAR-IoT framework.

### 3.10. Node-RED Workflows

Node-RED is a low-code programming tool that enables the integration of hardware devices, APIs, and online services in novel and innovative ways. It features a browser-based editor that facilitates the creation of flows, using a diverse range of nodes from the palette, which can be deployed to its runtime with a single click [37]. Node-RED can generate many flow combinations for various use cases, as demonstrated by an example use case of alerting in Figure 5. The Redis-In or MQTT node (based on the pub–sub broker in use) is required to read data from RADAR-IoT. The example in Figure 5 illustrates an alert workflow that triggers an email if no data are received within a specified time window. This feature enables administrators to investigate potential issues with data collection. Another flow demonstrated is reading audio from the MQTT broker, performing feature extraction, and then uploading it to the cloud.

This low-code solution using nodes and flows makes the framework’s configuration highly extensible, and, with relatively low effort, a wide variety of use cases is possible.

### 3.11. Methods of Deployment

The RADAR-IoT platform is modular, comprising three separate components that need to be deployed. (as shown in Figure 4):The Python module that connects to sensors and handles I/O. This module also validates and converts the data, using Avro schemas [46] before publishing to the pub–sub broker, as depicted by the blue components in Figure 4.The publish–subscribe broker (current options include Redis [35], MQTT [36]) shown by the dashed pink lines in the middle in Figure 4.The data consumer module subscribes to the sensor data, uploads data to the cloud, and performs limited on-device analytics. These are shown by the crimson components in Figure 4.

Packaging the framework as Docker images allows us to use orchestration frameworks like Nebula [50] to remotely deploy, maintain, and upgrade large fleets of IoT hub devices. The RADAR-IoT framework supports visualization tools like Grafana and the RADAR-base platform as backends. Most cloud IoT platforms support publish–subscribe brokers, allowing RADAR-IoT to integrate into multiple cloud backends easily. These types of extensions are supported through the simple no-code Node-RED configuration flows (see below).

### 3.12. Deployment Configurations

The RADAR-IoT framework’s modular architecture allows for deployment in various configurations, to meet diverse use cases. Below are some typical patterns.

#### 3.12.1. Multiple Primary Devices

As Figure 6a shows, this configuration allows one or multiple autonomous IoT gateway devices running RADAR-IoT to gather data from sensor sets. These devices operate independently and do not communicate with each other.

#### 3.12.2. Primary and Subordinates

The proposed configuration, as illustrated in Figure 6b, involves the utilization of a primary–subordinate architecture within the RADAR-IoT framework. A single device hosting all framework components is the primary gateway in this setup. Multiple subordinate devices equipped with sensors are responsible for interfacing with and collecting sensor data. These subordinate devices transmit the collected data to the primary device. The primary device can also directly interface with and collect data from the sensors if needed. However, only the primary device interacts with data uploading and consuming services, like dashboards and cloud platforms.

This configuration offers several advantages, especially when deploying multiple independent IoT gateway devices is not feasible or resource constraints exist. The primary–subordinate architecture optimizes data collection and management while minimizing resource utilization. This approach ensures efficient and centralized data processing while accommodating limitations regarding available resources or deployment feasibility. This architectural approach may be more susceptible to failures, as the primary device can represent a single point of failure at the edge. If the primary device malfunctions or becomes unavailable, it could cause a complete outage of all sensor data collection, potentially resulting in significant data loss or interruption of critical monitoring functions.

#### 3.12.3. Local Network Only

While deployment configurations (a) and (b) typically utilize a remote or cloud-hosted RADAR-base platform, it is also possible to deploy RADAR-base on the local network servers and so satisfy criteria for local deployment. This configuration precludes internet access, typically designated for in-hospitals where transmitting data outside the internal network is not desired. Since all RADAR-IoT components and associated external services are open-source, they can be deployed on local network computers/servers and operate effectively in a completely isolated “air-gapped” environment. The example configuration depicted in Figure 6c comprises the RADAR-base platform, the Grafana dashboard for visualization, and the AI/ML data consumer for anomaly detection, all supported by RADAR-IoT.

### 3.13. Methods for Management, Maintenance, and Monitoring

The management and orchestration of services on an IoT platform is an essential aspect of non-trivial deployments, as the deployment may need to scale to hundreds of thousands of devices. A typical scenario is having many devices and needing to update the version of the sensor interface service on all of them; doing this manually on each of those (even remotely) is not feasible. RADAR-IoT addresses this through automation, using external open-source management, maintenance, and monitoring solutions. These options presently include the following:Management and maintenance: Nebula [50] (an orchestration tool for IoT devices that use Docker-based services) and Dataplicity [51] (a remote IoT device management service)Monitoring: Node-RED [37] + Grafana [34] (providing inter- and intra-device monitoring and alerting) and Netdata [52] (professional system monitoring)

## 4. Proof of Concept (PoC)

The proof of concept (PoC) demonstration of the RADAR-IoT framework aimed to evaluate the framework in real-world settings, confirm the reliability of the integrated framework and sensors, and collect sensor data under controlled test conditions. A Raspberry Pi 4B device [31] with four types of sensors was used to deploy the framework in both an office and a home environment. Data were collected over three months, with notable events documented for 14 days, and they were analyzed using a Grafana dashboard and the RADAR-base platform. The PoC is an evaluation of the framework and its integrations rather than of the hardware devices used.

### 4.1. Aims

The objectives of the proof of concept (PoC) demonstration of the RADAR-IoT framework are as follows:Evaluate and demonstrate the framework in a real-world setting.Establish that the integration framework and the sensors perform as expected.Collect real-world sensor data with control test conditions, to assess changes in state to a known perturbation.

### 4.2. Methods

For this study, we deployed the RADAR-IoT framework in two distinct settings: a shared office space and a home-based environment, using a Raspberry Pi 4B device (4 GB model) [31] equipped with four sensors—air quality, motion, temperature, humidity, and light—connected via the Grove Pi hat [41]. The 64-bit Raspbian OS (based on Debian Linux) was used to deploy the RADAR-IoT framework on the device. We deployed the framework components using Docker images, to ensure reproducible environments. The Docker images were based on Python 3.9 for the Python sensor interface module and Java Development Kit (JDK) version 11 for the Kotlin-based data consumer module. Redis Pub/Sub was the choice of publish–subscribe broker, deployed using the official Redis Docker image. The Docker Compose files used in this proof of concept are provided in the GitHub repository [53]. The configurations used for various sensors and destination targets in the PoC are provided in the project wiki [54]. This information can be used to reproduce the exact environment and settings used for the proof of concept.

Data were collected over three months, during which a diary was kept for 14 days, to document any positive control events in the home-based setting, such as coughing, sneezing, opening a window, physical activity in the room, and turning heating on/off. We configured the RADAR-IoT framework to upload the data to a real-time Grafana dashboard for visualization and the RADAR-base platform for historical analysis.

### 4.3. Results

The first deployment of the framework was on a Raspberry Pi 4B [31] with a GrovePi Plus hat [41] and three plugged-in sensors installed in a shared office space. Figure 7 shows the dashboard screen for a week of data collected from the above office deployment.

The second deployment was done in a home-based environment. After pre-processing the data (excluding outliers) and taking the days with at least one “open-window” diary event, the correlation between the positive diary event of “open-window” with the “air-quality” sensor values was found to be, on average, −0.61 using the Spearman correlation coefficient [55]. Opening the window improved the room’s air quality and decreased the air pollutants (as measured by the sensor), resulting in a negative correlation.

Figure 8 presents the light sensor data collected over a week of monitoring. The plot shows expected patterns of light values, with higher values recorded in the morning when the sun rose and the room received natural light. As the sun started to set, there was a gradual decrease in the amount of light recorded by the sensor. When the artificial light in the room was turned on, the values jumped steeply, to form a second smaller peak (to the right side of the sunlight peak). The red line in the plot marks the local time of 10:30 p.m., which is approximately when the room user would go to sleep and turn off the light, resulting in the light sensor values dropping to near 0.

The data collection apparatus was left on for approximately three months, to measure the completion of data. Monitoring of the setup was limited to biweekly checks, with only one issue reported during the monitoring period, due to a change in the home’s Wi-Fi network. Table 2 presents the completion metrics for the sensors over the three months, providing valuable insights into the reliability and performance of the data collection system.

## 5. Discussion

As shown in Figure 9, using the RADAR-IoT framework with the RADAR-base mHealth platform provides a 360-degree view into a user’s health and environment, combining static IoT sensors dependent on an edge device and ambulatory wearable devices, providing the possibility of combining rich information from wearable devices, mobile phone sensors and interactions, questionnaires and tasks, and environmental sensors (such as indoor air quality, temperature, humidity, and toxic gas levels).

Compared to other IoT platforms in Table 1, the RADAR-IoT framework offers several unique advantages and innovations that make it stand out:360-Degree View: Unlike many other platforms, RADAR-IoT provides a comprehensive 360-degree view into the user’s health and environment, as mentioned above. This rich information set provides a more holistic understanding of the user’s context.Open-Source and Versatile: RADAR-IoT is an open-source solution, which means it can be freely used, modified, and distributed. It can be used in any use case and supports most protocols, making it more versatile than many other proprietary platforms that are limited in their use cases and protocol support.Edge Support: One of the critical advantages of RADAR-IoT is its edge support. It provides dashboards, data processing, and analysis capabilities at the edge, which is uncommon in many other platforms. These capabilities enable more efficient data processing and real-time insights.Cloud Support for Health Data: RADAR-IoT offers cloud support for health data, using the RADAR-base platform [9]. This feature is unique to RADAR-IoT and is not found in many other platforms, especially those without health-specific features.No-Code Extensions: RADAR-IoT supports no-code extensions, using Node-RED wiring [37], allowing easy customization and extension of the platform without the need for extensive coding, making it more accessible to non-technical users.

In comparison, many other platforms listed in the table either lack health-specific features, have limited cloud or edge support, are not open-source, or do not provide a 360-degree view of a user’s health and environment. These factors highlight the advantages and innovations of RADAR-IoT in the IoT for Health landscape. While RADAR-IoT is still in development and has a few limitations, such as a lack of ease during setup and no support for wireless sensors, its ongoing development means these issues are being addressed and improved.

We now discuss some potential application areas combining RADAR-IoT with the existing ambulatory data collection capabilities of RADAR-base. Managing infection control can be challenging in hospital wards. Exposure monitoring could be performed by tracking room visits and using air quality sensors to measure room particulate and air humidity levels, to give a readout of air quality status or airflow dynamics through a building. Additionally, we could combine cough counting with microphones and use edge device classification analysis of the audio.

Similarly, we could enable continuous passive monitoring of coughs, sneezing, and snoring (including apnea and hypopnea) in home settings with a similar setup, using a microphone, air quality sensor, wearable devices, and static bed mattress sensors. Combining IoT with ambulatory sensors is attractive, especially in cases where the environment contributes directly to the disease and its symptoms. For instance, in the case of chronic pulmonary disorders (e.g., COPD or asthma), poor air quality or allergens can have an adverse effect on the user’s health, including increased duration and intensity of coughing and wheezing. Poor air quality can be an essential factor in the relapse of depression and anxiety [60]. Similar studies show air quality improved after introducing the London ultra-low emission zones (ULEZ) [61]. These were introduced due to high air pollution in the city causing health issues for residents [62].

Another example use case is for monitoring conditions where the participant’s impaired motor control or cognitive ability affects their capacity for everyday living function: for example, in dementia, such as Alzheimer’s disease [63]. In such cases, the RADAR-IoT framework can capture certain environmental events, which could help in understanding the progression of disease or loss of activities of daily living, e.g., using door or motion sensors to detect when a participant opens a door or cupboard—using a range of different sensing capabilities to monitor and build up a picture of home-based utility while simultaneously complementing this with wearable sensors, to provide insight into their behavior and physiology. With an increasingly elderly population, improved in-community care for patient populations will be a significant requirement; this type of continuous, passive monitoring and risk assessment or alerting on detecting critical events could be incorporated into and augment a future care system.

The RADAR-IoT framework could also be implemented in the clinical environment, to trace the conversation between a patient and a doctor. Using audio sensors, it could record these events and further analysis that would sufficiently provide insight into the patient’s health status or even a prognosis on how effective specific treatments are. This would be particularly useful in a mental health clinic, wherein subtle speech characteristics may reveal a patient’s mental state. It could pick up on tone, speed, and volume of speech, even detecting feelings of distress or confusion, allowing real-time feedback to health professionals so that they could adjust their approach or treatment plan as needed.

The RADAR-IoT framework further permits monitoring a patient’s physical health in a non-intrusive manner. To elaborate, continuous follow-up of vital signs that include heart rate, blood pressure, and oxygen level could be managed with wearable devices. These data could be linked to environmental sensor data capturing temperature and humidity levels to build an all-rounded view of patients’ health and environment. This could be very useful in chronic conditions, such as diabetes or heart disease, whereby continuous monitoring would give indicators of an oncoming health crisis.

## 6. Conclusions

RADAR-IoT takes inspiration from existing work in IoT and seeks to address several limitations with current IoT gateway frameworks for health research. The limitations encompass source code availability, sensor types and I/O protocols, interoperability and scalability, cloud platform support, and edge processing [7,8,29,64].

The design and proof-of-concept deployment highlight some critical areas where RADAR-IoT differentiates itself from related systems. Firstly, the system is device-, sensor-, and programming language-agnostic for interoperability. The open-source nature of RADAR-IoT makes it readily extensible, and the modular design allows for ease of composition. Lastly, it supports a wide range of IoT input/output protocols and multiple data sinks, including but not limited to the RADAR-base mHealth platform, on-device data processing, and Grafana dashboards [34]. On the security and privacy side, RADAR-IoT supports industry-leading security and privacy, using standards such as identity management, OAuth 2.0 [42], encryption in transit and at rest, pseudonymization, and data typing and schematization. Integration with RADAR-base [9] provides a well-established open-source mHealth cloud backend for data collection, aggregation, transformation, compute-intensive analytics, and a combination of different types of data sources, like wearables, IoT sensors, mobile apps, and eCRFs.

We have demonstrated a proof of concept using RADAR-IoT to flexibly connect a range of IoT sensors and how these data could be collected and used with various third-party integrations. Incorporating static IoT sensors with the RADAR-base platform provides a comprehensive view of the patient’s health and environment, allowing for a more holistic understanding of how these factors interact with and impact patient outcomes.

RADAR-IoT provides a reference architecture as a solid foundation for implementing IoT in health research. Its open-source data collection capabilities are crucial for the future of research, healthcare automation, and responsive infrastructure. Despite its novelty and effectiveness in real-world settings, the platform has limited developer documentation, unproven usefulness in a health research area, complexity during initial setup, and limited remote management and control support.

Looking ahead, we plan to integrate RADAR-IoT with a broader range of sensor types. This expansion will enable the system to collect a more diverse range of data, enhancing its ability to provide a comprehensive view of the user’s health and environment. We also aim to enhance RADAR-IoT’s data analysis capabilities by incorporating more sophisticated algorithms for interpreting sensor data and deriving insights. This enhancement will improve the system’s ability to detect patterns and trends in the data, which could be vital for early identification of health issues. In addition to these improvements, we plan to deploy RADAR-IoT more broadly in clinical settings. This broader deployment will allow us to gather more data and better understand its performance and effectiveness in a real-world health environment. We also plan to bundle the software framework and the hardware device as a solution for ease of use. This solution could be preconfigured with sensors, e-sims for mobile network connectivity, portable power sources, and predefined software components based on disease types for a plug-and-play solution.

To fully demonstrate the practical value of the framework, we plan to conduct further pilot studies that evaluate its performance in real-world health research use cases. These studies will involve a detailed analysis of data collected from wearables, apps, and IoT sensors and will aim to provide concrete evidence of the framework’s effectiveness and combined utility in improving patient outcomes. As part of our future work, we will scientifically evaluate quantitative performance metrics, such as data throughput, latency, and energy consumption for the deployed sensors. This evaluation will help us identify areas where the system’s performance could be improved, ensuring that RADAR-IoT continues to meet the needs of the health domain.

## Figures and Tables

**Figure 1 sensors-24-04614-f001:**
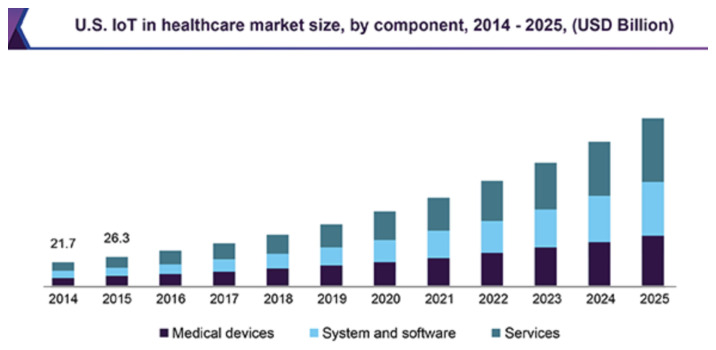
The market trend of the internet of things (IoT) in health [5].

**Figure 2 sensors-24-04614-f002:**
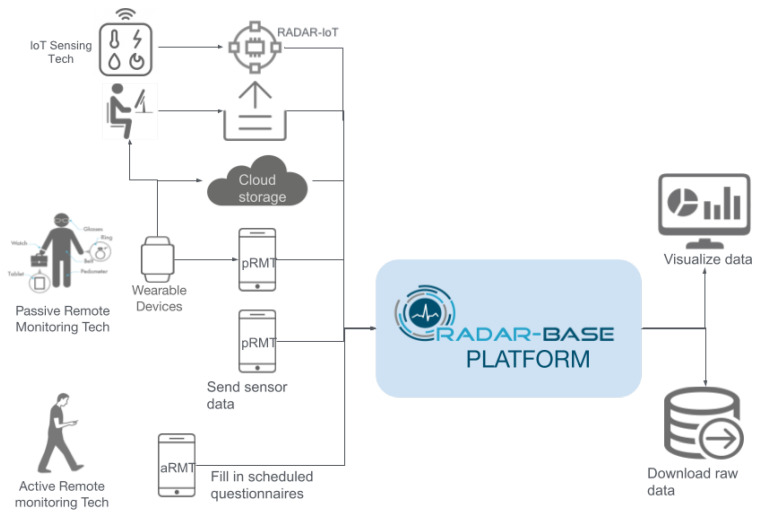
An. overview of the RADAR-base platform [9] and integration of RADAR-IoT.

**Figure 3 sensors-24-04614-f003:**
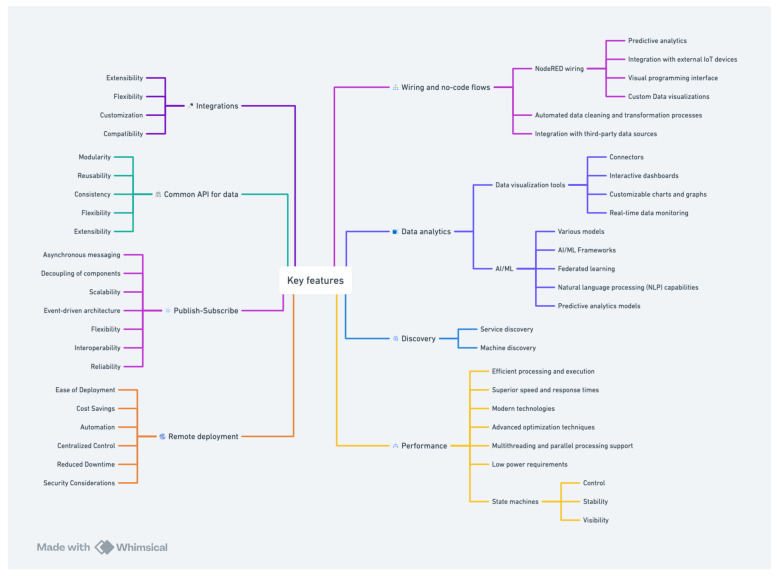
RADAR-IoT design requirements.

**Figure 4 sensors-24-04614-f004:**
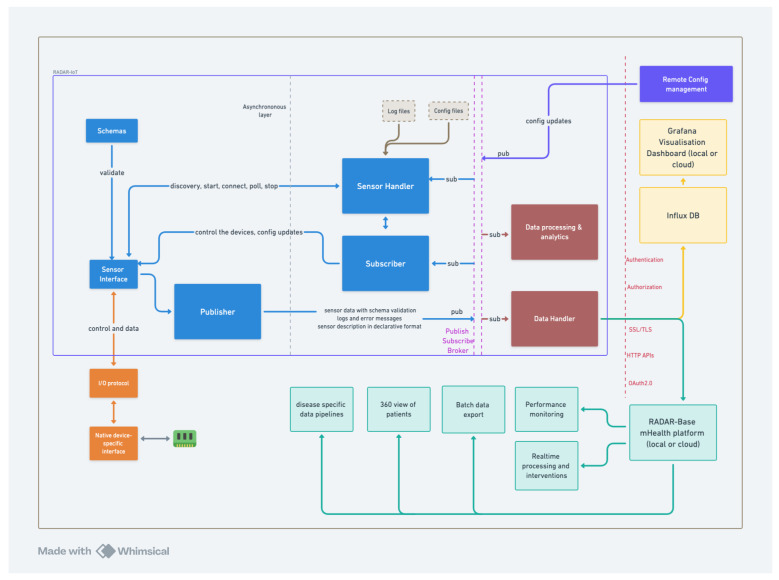
Architecture overview of RADAR-IoT.

**Figure 5 sensors-24-04614-f005:**
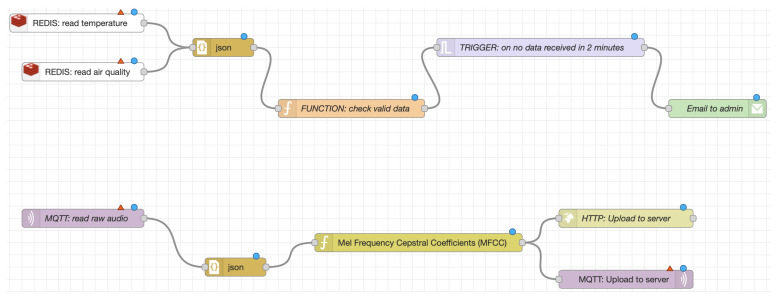
Example Node-RED workflows augmenting RADAR-IoT.

**Figure 6 sensors-24-04614-f006:**
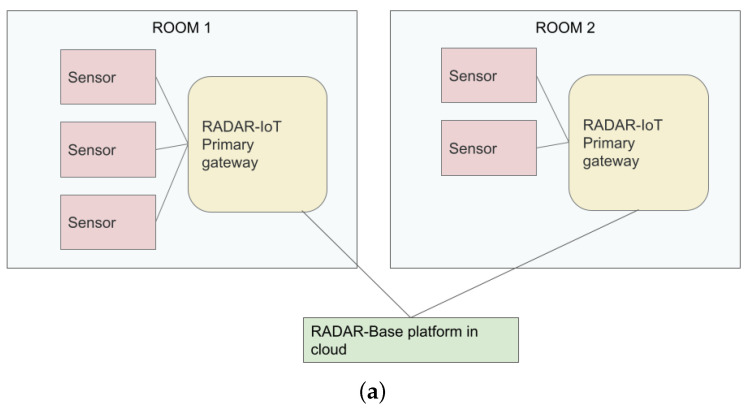
RADAR-IoT deployment configurations. (**a**) RADAR-IoT simple deployment configuration. (**b**) RADAR-IoT deployment in primary–subordinate configuration. (**c**) RADAR-IoT deployment configuration for local network (no internet access required).

**Figure 7 sensors-24-04614-f007:**
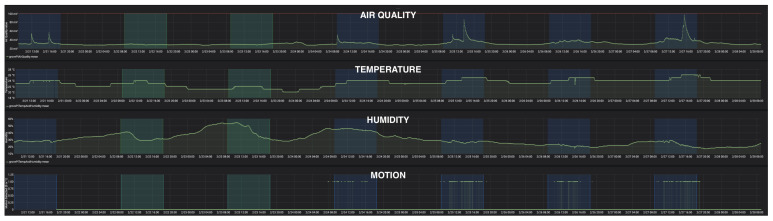
Real-time dashboard. The sensors are air quality, temperature, humidity, and motion, from top to bottom. The green areas are times of day (9:00–18:00) during the weekends (holiday), while the blue areas are times of day (9:00–18:00) during the weekdays. As expected, the values differ between the weekdays and weekends for almost all the data, due to variations in room population. For instance, weekdays have higher motion, air quality, and temperature, since people work in rooms with lower humidity than on weekends.

**Figure 8 sensors-24-04614-f008:**
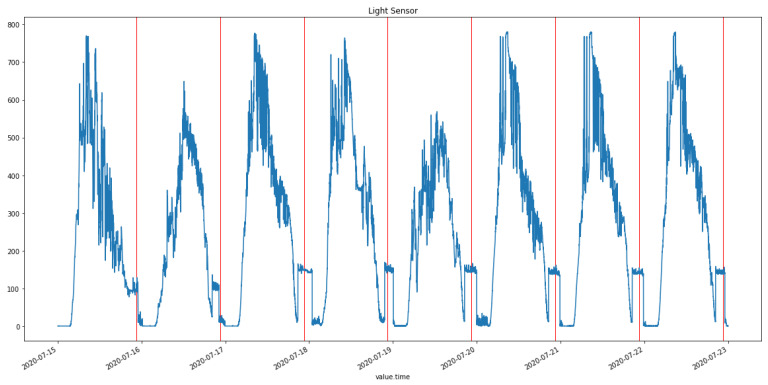
Light sensor data. Intraday changes can be seen corresponding to sunrise and sunset.

**Figure 9 sensors-24-04614-f009:**
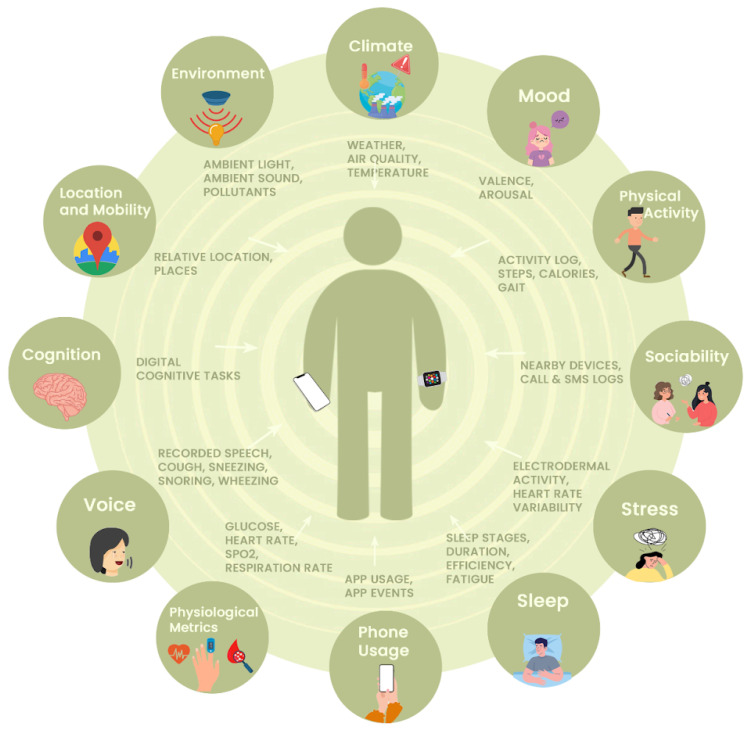
RADAR-IoT and RADAR-base platforms enable the capture of potential modalities for health research.

**Table 1 sensors-24-04614-t001:** IoT gateway frameworks comparison.

No.	Name	Open-Source?	Usage	Protocols	Advantages	Disadvantages
1	ESF IoT Edge Framework (based on Eclipse Kura) [18]	No	Commercial or Industrial	General industrial protocol support (like GPIO).	Multiple clouds supported; visual wiring for data flows.	No health-specific features.
2	Eclipse Kura [19]	Yes	Commercial or Industrial	Most industrial protocols (like GPIO) and MQTT.	Extensible using Java, with visual wiring for data flows.	Limited cloud support, and hard to set up. No health-specific features.
3	A Smart Gateway Framework for IoT Services [20]	No	Industrial	Variable.	Good support and extensibility with ontologies, context manager, and device catalog.	Limited data processing using Rules; only supports Thingspeak; requires at-edge HTTP server that needs to open ports. No health-specific features.
4	ubiworx IoT Gateway Software Framework [21]	No	Commercial or Industrial	Most protocols supported.	Good connections and protocol support for clouds.	Extending can be hard with Lua scripting. No health-specific features.
5	A Novel Cognitive IoT Gateway Framework [22]	No, only a theoretical framework	Any use case	Unclear but theoretically could be added in the cognitive framework.	Computation and analytics on data from sensors based on the rules set.	Added complexity for extending. No clear support for protocols or the cloud. No health-specific features.
6	DeviceHive [23]	Yes	Any use case	Limited support.	Good support for features at edge.	Poor cloud support. No edge computing or data processing. No health-specific features.
7	ThingSpeak [24]	Yes	Any use case	None; primarily a cloud-based platform.	Ease of setup and use.	Only self (Thingspeak) cloud support; no edge computing (only cloud-based analytics). No health-specific features.
8	Thinger.io [25]	Yes	Any use case	None; primarily a cloud platform.	Node-RED flows wiring and real-time display and triggers.	Cloud-based only. No health-specific features.
9	Zetta [26]	Yes	Any use case	Limited support.	Good extensibility of devices at edge with reactive paradigm.	No edge computing. No health-specific features.
10	Open Remote [27]	Yes	Industrial	Most common protocols supported.	Good support for edge computing with data analytics, rules, and automation.	Only self (Open Remote) cloud supported. No health-specific features.
11	ThingsBoard Gateway [28]	Yes	Any use case	Most protocols supported.	Ease of use and setup; supports common protocols for easy integration.	Only ThingsBoard cloud supported. Limited edge computing. No health-specific features.

**Table 2 sensors-24-04614-t002:** Completion of data from home-based monitoring.

Sensor	Description	Percent Completion (%)	Sampling Interval (Seconds)
Air Quality [56]	This sensor is a comprehensive indoor air monitor, responsive to a variety of harmful gases, including carbon monoxide, alcohol, acetone, thinner, and formaldehyde.	90.94	5
Temperature and Humidity [57]	This sensor is a pre-calibrated digital device for measuring temperature via a negative temperature coefficient (NTC) thermistor and relative humidity through a unique capacitive element.	91.46	10
Motion [58]	This sensor detects motion, specifically human movement within its range, and outputs HIGH on its signal pin when motion is detected.	83.58	1
Light [59]	This sensor uses a photo-resistor to detect light intensity, producing an analog output signal that increases with brightness, facilitated by an on-board LM358 OpAmp chip.	86.59	30

## Data Availability

All the data from the proof of concept (including the events diary data) are available publicly online [65]. The data are pseudonymized according to the policies and processes of the RADAR-base platform, which provides pseudonymized sensor data.

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
