# Peer review of "RADAR-IoT: An Open-Source, Interoperable, and Extensible IoT Gateway Framework for Health Research"

_sensors, 2024, doi:10.3390/s24144614_

Round 1
Reviewer 1 Report
Comments and Suggestions for Authors
I would like to thank the authors for their effort and work. As per my option, the manuscript required extreme editing before it can be accepted for publication. It is not easy to follow. Specifically the structure of the manuscript is not well organized, as a result it is not easy to understand.
Abstract
In abstract it is written that ‘perform on-device data analysis’, How much of it should be possible? As the edge gateway should possibly be the place where the data analysis should take place because IoT devices are resource constraints.
My suggestion for improvement of the manuscript.
-
Introduction : The introduction section started directly with subsection 1.1. Internet of Things, here an introduction a compiled version of the subsection is expected and the research problem should be introduced. Please add a short introduction to the domain and problem which will end with two important paragraphs: 1) The main contributions in the current article (the content of current section 1.4. Design Requirements can be slightly modified and added within this part), 2) The structure of the manuscript (as you can see an example at the last two paragraphs of the introduction section here https://www.mdpi.com/2227-9032/10/10/1993). The first two paragraphs of current 1.1. Internet of Things section doesn’t contain any citation, this should include several citations to validate the arguments presented by the authors. The current first paragraph of introduction can be included here with more clear and detailed descriptions on motivations and citations.
-
Background: The background section can include the current 1.1. Internet of Things (starting from the 2nd paragraph) and presented as 2.1. Internet of Things within 2. Background section where different components of the research can be explained separately. Please include 3-4 lines within the 2. Background section before starting 2.1. Internet of Things. The current 1.2. RADAR-Base should be included as 2.2. RADAR-Base.
-
Related works: The motivation part from the current 1.3. Motivation and Related Work should go within the 1. Introduction section and the Related Work (including Table 1. IoT gateway frameworks comparison) should go here, but the row 12 should be included within the discussion section and can be described with sentences.
-
Proposed Architecture/Proposed Solution: Present your proposed model/artifact here (the current section 2 Architecture can be reused here). The current section 3. Deployment Configurations should possibly be adopted with the current 2.8. Methods of Deployment (may be 4.8. Methods of Deployment). Please include a modified version of Figure 1. Within this section to clearly present your contribution to the architecture/model.
-
Results: The current 4. PoC Demonstration section can be renamed and a few lines should be added before the subsection, introducing the approach adapted for data collection for validating the PoC.
-
Discussion: The current 5. Value Addition by RADAR-IoT section can be renamed and presented here.
-
Conclusion and Future work: The current 6. Conclusion can be reused here.
Comments on the Quality of English Language
it would be beneficial to add a few easy-to-understand sentences at the beginning of each section. Additionally, ensuring the coherence of the sections is crucial.
Author Response
Thank You for providing valuable feedback on the manuscript. Please find the detailed responses below and the corresponding revisions in the re-submitted files. We have incorporated all the changes requested and hope for your positive response.
Comment 1: Abstract: In abstract it is written that ‘perform on-device data analysis’, How much of it should be possible? As the edge gateway should possibly be the place where the data analysis should take place because IoT devices are resource constraints.
Response 1: We have updated “perform on-device data analysis” to “perform limited on-device data processing and analysis” in the manuscript [Line 10].
We have added a section (Section “3.9 Data Processing and Analysis”) in the manuscript to discuss this aspect of the framework in detail [Line 338].
We agree that advanced data analysis might not be feasible on resource-constrained edge devices. However, rudimentary analysis, such as feature extraction, should be possible with recent advancements. Please take a look at section 3.9 in the manuscript for details on this functionality.
My suggestion for improvement of the manuscript.
Comment 2: Introduction : The introduction section started directly with subsection 1.1. Internet of Things, here an introduction a compiled version of the subsection is expected and the research problem should be introduced. Please add a short introduction to the domain and problem which will end with two important paragraphs: 1) The main contributions in the current article (the content of current section 1.4. Design Requirements can be slightly modified and added within this part), 2) The structure of the manuscript (as you can see an example at the last two paragraphs of the introduction section here https://www.mdpi.com/2227-9032/10/10/1993). The first two paragraphs of current 1.1. The Internet of Things section doesn’t contain any citation, this should include several citations to validate the arguments presented by the authors. The current first paragraph of introduction can be included here with more clear and detailed descriptions on motivations and citations.
Response 2: We added an introductory paragraph summarizing the introduction section [Line 21]. We have added the two requested paragraphs to the introduction [Lines 48-65]. We have also added relevant citations to the requested two paragraphs [Lines 29-47].
Comment 3: Background: The background section can include the current 1.1. Internet of Things (starting from the 2nd paragraph) and presented as 2.1. Internet of Things within 2. Background section where different components of the research can be explained separately. Please include 3-4 lines within the 2. Background section before starting 2.1. Internet of Things. The current 1.2. RADAR-Base should be included as 2.2. RADAR-Base.
Response 3: We created a 2. Background section in the manuscript and moved the last two paragraphs of 1.1 Internet of Things to 2.1 Internet of Things [Line 98]. We also moved 1.2 RADAR-base to 2.2 RADAR-Base [Line 89] and added a small introduction at the start of the Background section [Lines 90-97].
Comment 4: Related works: The motivation part from the current 1.3. Motivation and Related Work should go within the 1. Introduction section and the Related Work (including Table 1. IoT gateway frameworks comparison) should go here, but the row 12 should be included within the discussion section and can be described with sentences.
Response 4: As requested, we have moved the Motivation to the Introduction section in the manuscript [Line 66]. We have moved row 12 from Table 1 to text in the Discussion section [Lines 489-495].
Comment 5: Proposed Architecture/Proposed Solution: Present your proposed model/artifact here (the current section 2 Architecture can be reused here). The current section 3. Deployment Configurations should possibly be adopted with the current 2.8. Methods of Deployment (may be 4.8. Methods of Deployment). Please include a modified version of Figure 1. Within this section to clearly present your contribution to the architecture/model.
Response 5: We have renamed this section "3. Proposed Solution" [Line 183] and moved the Deployment configurations below the "Methods of Deployment" to improve coherence [Section 3.12, Line 380].
We have included a modified version of the overview of the RADAR-Base platform Figure to show the integration of RADAR-IoT into the platform [Figure 2].
Comment 6: Results: The current 4. PoC Demonstration section can be renamed and a few lines should be added before the subsection, introducing the approach adapted for data collection for validating the PoC.
Response 6: We have renamed the section to “Proof-of-Concept (PoC)” [Section 4, Line 429]. We have also added a few lines as an introduction to this section [Lines 430-437].
Comment 7: Discussion: The current 5. Value Addition by RADAR-IoT section can be renamed and presented here.
Response 7: We have renamed this section to 5. Discussion [Line 479]. We have also added an image to depict the discussion points [Figure 9].
Comment 8: Conclusion and Future work: The current 6. Conclusion can be reused here.
Response 8: Ok, we have kept this section as it is [Line 526].
Comment 9: Comments on the Quality of English Language
it would be beneficial to add a few easy-to-understand sentences at the beginning of each section. Additionally, ensuring the coherence of the sections is crucial.
Response 9: We have included introduction paragraphs at the beginning of each section to make the text easier to read. Additionally, we have reorganized some parts of the text to enhance the overall flow of the manuscript. Furthermore, the manuscript has been carefully reviewed and edited by two native English speakers and tools like Grammarly to ensure the precision and accuracy of the English language.
Reviewer 2 Report
Comments and Suggestions for Authors
This paper introduces RADAR-IoT, an IoT edge framework designed for health research, emphasizing extensibility, modularity, and support for various sensors. RADAR-IoT enables real-time data collection, processing, and analysis, facilitating remote monitoring and just-in-time interventions in healthcare applications, ultimately aiming to improve health outcomes through continuous and remote data collection. The following are suggestions from the reviewer:
1. The abstract (Lines 1-12) provides a comprehensive overview of the research. However, it could benefit from a more structured format. Consider clearly separating the motivation, methods, results, and conclusions within the abstract to improve readability.
2. While the paper provides a good overview of the RADAR-IoT framework (Lines 125-145), some technical details are lacking. More information about the specific algorithms used for on-device data analysis and the exact hardware specifications for the proof-of-concept implementations would be valuable.
3. The proof-of-concept section (Lines 337-382) demonstrates the framework's capabilities but lacks quantitative performance metrics. Include metrics such as data throughput, latency, and energy consumption for the deployed sensors to provide a clearer picture of the system's performance.
4. The manuscript mentions the importance of security and privacy (Lines 228-244) but lacks specific details on how these aspects are implemented. Provide a more detailed explanation of the security protocols and privacy measures, including any encryption standards and pseudonymization techniques used.
5. While the paper provides a good overview of the RADAR-IoT framework, some technical details are lacking. For instance, in lines 125-145, the manuscript mentions the integration of various sensors but does not specify the algorithms used for data processing. Detailed descriptions of these algorithms and their computational requirements would enhance understanding. Additionally, exact hardware specifications, including model numbers and configurations, should be provided to allow reproducibility of the experiments.
6. The paper would benefit from a section discussing the user experience, particularly for researchers and healthcare professionals using the system. Include feedback from any pilot users or studies that highlight the system's ease of use and any identified areas for improvement.
7. The manuscript briefly touches on future improvements (Lines 423-461). Expanding this section to include more specific plans, such as integration with additional sensor types, improvements in data analysis capabilities, or broader deployment in clinical settings, would strengthen the conclusion.
8. Ensure that all references are correctly formatted according to the journal's guidelines. Additionally, a thorough proofreading to correct any grammatical errors and improve sentence structure will enhance the overall readability of the manuscript.
Error 1: Line 1: "Internet of Things (IoT) sensors offer a wide range of sensing capabilities, many of which have potential for health applications." - "have potential for" should be "have the potential for".
Error 2: Line 34: "advanced artificial intelligence has becomes increasingly feasible" - "has becomes" should be "has become".
The manuscript "RADAR-IoT" is promising but has critical issues: an unstructured abstract, insufficient technical details, missing quantitative metrics, inadequate security and privacy explanations, and no user feedback. Additionally, the future work section needs expansion, and there are grammatical errors and formatting issues. I recommend rejecting the manuscript in its current form. Substantial revisions are needed before resubmission.
Comments on the Quality of English Language
Moderate editing of English language required
Author Response
Thank You for providing valuable feedback on the manuscript. Please find the detailed responses below and the corresponding revisions in the re-submitted files. We have incorporated all the changes requested and hope for your positive response.
Comment 1: The abstract (Lines 1-12) provides a comprehensive overview of the research. However, it could benefit from a more structured format. Consider clearly separating the motivation, methods, results, and conclusions within the abstract to improve readability.
Response 1: OK, we have structured the abstract now and separated it into suggested sections [Lines 1-18].
Comment 2: While the paper provides a good overview of the RADAR-IoT framework (Lines 125-145), some technical details are lacking. More information about the specific algorithms used for on-device data analysis and the exact hardware specifications for the proof-of-concept implementations would be valuable.
Response 2: We have also added a section “3.9 Data processing and analysis” [Line 338] to the manuscript.
Table 2 includes a basic description of the hardware specifications of the sensors used in the PoC. The detailed hardware specs can be viewed by looking at the cited sources in Table 2.
Comment 3: The proof-of-concept section (Lines 337-382) demonstrates the framework's capabilities but lacks quantitative performance metrics. Include metrics such as data throughput, latency, and energy consumption for the deployed sensors to provide a clearer picture of the system's performance.
Response 3: We have also added more technical details on how the framework is optimzed for performance in a new section, “3.7 Performance” [Line 299]. However, since this is just a software framework, we did not quantify these metrics as they are heavily dependent on the hardware used and the types of input/output interfaces. As a methods paper for the gateway framework, we think this is out of the scope of this paper as the hardware will vary with each use case in which RADAR-IoT is used. However, we recognise it as an important future work and have added it to the manuscript.
The PoC was an exemplar of our generic framework aimed at testing the apparatus's feasibility in a real-world setting. It did not aim to evaluate performance; however, we included sensor data completion rates in the PoC to demonstrate the system's data collection efficacy.
Comment 4: The manuscript mentions the importance of security and privacy (Lines 228-244) but lacks specific details on how these aspects are implemented. Provide a more detailed explanation of the security protocols and privacy measures, including any encryption standards and pseudonymization techniques used.
Response 4: We have added more detailed security and privacy measures to the manuscript, including specific techniques [Lines 264-298].
Comment 5: While the paper provides a good overview of the RADAR-IoT framework, some technical details are lacking. For instance, in lines 125-145, the manuscript mentions the integration of various sensors but does not specify the algorithms used for data processing. Detailed descriptions of these algorithms and their computational requirements would enhance understanding. Additionally, exact hardware specifications, including model numbers and configurations, should be provided to allow reproducibility of the experiments.
Response 5: We have added a section (Section “3.9 Data Processing and Analysis”) in the Manuscript to provide information on data processing [Line 338].
We have added more technical details about the framework in a new section, “3.7 Performance” [Line 299].
Table 2 includes a basic description of the hardware specifications of the sensors used in the PoC. The detailed hardware specs can be viewed by looking at the cited sources in Table 2.
Comment 6: The paper would benefit from a section discussing the user experience, particularly for researchers and healthcare professionals using the system. Include feedback from any pilot users or studies that highlight the system's ease of use and any identified areas for improvement.
Response 6: This is not applicable as the system has not been used in any studies yet and is out of scope for this methods paper. However, there are plans to use the framework to monitor patients and their environments inside hospitals in London.
Comment 7: The manuscript briefly touches on future improvements (Lines 423-461). Expanding this section to include more specific plans, such as integration with additional sensor types, improvements in data analysis capabilities, or broader deployment in clinical settings, would strengthen the conclusion.
Response 7: We have added more details on the planned future work in the last two paragraphs of the “6. Conclusion” section [Lines 555-576]. We have also added the “5. Discussion” section, discussing the potential research and clinical use cases where RADAR-IoT can be applied [Lines 496-525].
Comment 8: Ensure that all references are correctly formatted according to the journal's guidelines. Additionally, a thorough proofreading to correct any grammatical errors and improve sentence structure will enhance the overall readability of the manuscript.
Response 8: Since we used the journal's template, we assume the references' format is correct.
We have included introduction paragraphs at the beginning of each section to make the text easier to read. Additionally, we have reorganized some parts of the text to enhance the overall flow of the manuscript. Furthermore, the manuscript has been carefully reviewed and edited by two native English speakers and tools like Grammarly to ensure the precision and accuracy of the English language.
Comment 9: Error 1: Line 1: "Internet of Things (IoT) sensors offer a wide range of sensing capabilities, many of which have potential for health applications." - "have potential for" should be "have the potential for".
Response 9: We have fixed the error and updated the text in the manuscript [Line 1].
Comment 10: Error 2: Line 34: "advanced artificial intelligence has becomes increasingly feasible" - "has becomes" should be "has become".
Response 10: We have fixed the error and reworded the text in the manuscript [Lines 45-47].
Comment 11: The manuscript "RADAR-IoT" is promising but has critical issues: an unstructured abstract, insufficient technical details, missing quantitative metrics, inadequate security and privacy explanations, and no user feedback. Additionally, the future work section needs expansion, and there are grammatical errors and formatting issues. I recommend rejecting the manuscript in its current form. Substantial revisions are needed before resubmission.
Response 11: Thank you for the suggestions. We have substantially revised the manuscript and addressed all the points raised here. We hope for a positive response.
Comment 12: Comments on the Quality of English Language
Moderate editing of English language required
Response 12: We have included introduction paragraphs at the beginning of each section to make the text easier to read. Additionally, we have reorganized some parts of the text to enhance the overall flow of the manuscript. Furthermore, the manuscript has been carefully reviewed and edited by two native English speakers and tools like Grammarly to ensure the precision and accuracy of the English language.
Round 2
Reviewer 2 Report
Comments and Suggestions for Authors
Thank you for addressing the previous review comments and submitting a revised manuscript. The revised manuscript shows significant improvements in several areas, including technical details, performance metrics, security and privacy, user experience, and future plans. The structure is clearer, and the content is well-organized with smooth transitions between sections. Overall, the study is of high quality and meets the publication standards. However, there are a few minor revisions that need to be addressed:
1. Rewrite the abstract as a single paragraph, avoiding explicit separation of sections. This will improve readability and flow.
2. Expand the comparative analysis of existing solutions to better highlight the advantages and innovations of RADAR-IoT.
3. Provide more details on the data analysis methods, particularly the rationale behind the chosen algorithms and models.
4. Add more details about the experimental setup in the Proof-of-Concept section, such as specific configurations and descriptions of the test environments. This will aid reproducibility.
5. Include more qualitative feedback from actual users in the discussion section to complement the quantitative data. This will provide a more comprehensive view of the system's practical applications.
The revised manuscript shows notable improvements and is close to meeting the high standards for publication. The suggested minor revisions will further enhance the clarity and completeness of the study. I recommend accepting the paper after these minor revisions to correct methodological errors and improve text editing.
Comments on the Quality of English Language
Minor editing of English language required
Author Response
Thank You for providing valuable feedback on the manuscript. Please find the detailed responses below and the corresponding revisions in the re-submitted files. We have incorporated all the changes requested and hope for your positive response.
Comment 1: Rewrite the abstract as a single paragraph, avoiding explicit separation of sections. This will improve readability and flow.
Response 1: We have updated the abstract as a single paragraph without explicit sections [Lines 1-17]
Comment 2: Expand the comparative analysis of existing solutions to better highlight the advantages and innovations of RADAR-IoT.
Response 2: We have expanded the comparative analysis in the “5 Discussion” section [Lines 518-543]
Comment 3: Provide more details on the data analysis methods, particularly the rationale behind the chosen algorithms and models.
Response 3: We have added more details in the section “3.9. Data Processing and Analysis” including rationale for supported methods [Lines 333-370].
Comment 4: Add more details about the experimental setup in the Proof-of-Concept section, such as specific configurations and descriptions of the test environments. This will aid reproducibility.
Response 4: We have added more details about the software setup and configurations used in the proof-of-concept. We have also specified how to reproduce the experiments [Lines 469-481].
Comment 5: Include more qualitative feedback from actual users in the discussion section to complement the quantitative data. This will provide a more comprehensive view of the system's practical applications.
Response 5: Providing qualitative feedback from actual users is outside the scope of this paper as the developed solution has yet to be used in a real-world study. However, there are already plans to use it to continuously monitor cough for patients with lung disorders and a psychosis study to record speech in the clinic during patient visits. We will add qualitative results when we publish results for these studies in the future.
We have added a comprehensive account of the system’s potential practical applications in the “5 Discussion” section of the manuscript [Lines 544-588].